# DDM-HSA: Dual Deterministic Model-Based Heart Sound Analysis for Daily Life Monitoring

**DOI:** 10.3390/s23052423

**Published:** 2023-02-22

**Authors:** Miran Lee, Qun Wei, Soomin Lee, Heejoon Park

**Affiliations:** 1Department of Computer and Information Engineering, Daegu University, Kyeongsan 38453, Republic of Korea; 2Department of Biomedical Engineering, School of Medicine, Keimyung University, Daegu 42601, Republic of Korea; 3Clairaudience Company Limited, Daegu 42403, Republic of Korea; 4Department of Biomedical Engineering, Graduate School of Medicine, Keimyung University, Daegu 42601, Republic of Korea

**Keywords:** photoplethysmogram, phonocardiograms, daily life monitoring, heart sound, vascular transit time

## Abstract

A sudden cardiac event in patients with heart disease can lead to a heart attack in extreme cases. Therefore, prompt interventions for the particular heart situation and periodic monitoring are critical. This study focuses on a heart sound analysis method that can be monitored daily using multimodal signals acquired with wearable devices. The dual deterministic model-based heart sound analysis is designed in a parallel structure that uses two bio-signals (PCG and PPG signals) related to the heartbeat, enabling more accurate heart sound identification. The experimental results show promising performance of the proposed Model III (DDM-HSA with window and envelope filter), which had the highest performance, and S1 and S2 showed average accuracy (unit: %) of 95.39 (±2.14) and 92.55 (±3.74), respectively. The findings of this study are anticipated to provide improved technology to detect heart sounds and analyze cardiac activities using only bio-signals that can be measured using wearable devices in a mobile environment.

## 1. Introduction

Sudden cardiac events in patients with heart diseases or the occurrence of a particular heart event in healthy individuals may lead to a heart attack in extreme cases [1,2,3]. Therefore, prompt interventions depending on the particular heart event and periodic monitoring are crucial. Furthermore, among various heart diseases or events, arrhythmia is a precursory symptom that can influence the signs of abnormal cardiac symptoms, such as aortic stenosis and aortic insufficiency. In addition, because arrhythmia is a factor that induces stroke, cerebral infarction, and acute myocardial infarction, which are the leading causes of sudden death, studies to identify risk factors through continuous observation and monitoring of heart conditions in daily life have increased.

Among the identification methods of cardiac risk factors through observation of heart condition, coronary angiography (CAG) is the most accurate and precise standard examination method. This method involves injecting a contrast agent by inserting a thin tube with a diameter of 2 to 3 mm into the arterial blood vessel. However, subjects undergoing CAG cannot on the day of the examination, and other inconveniences exist, such as behavioral restrictions for a certain period after the examination. In addition, CAG is an invasive examination method [4]; therefore, this method poses an economic burden, and performing fast and convenient tests using this method in daily life is impossible. Therefore, an algorithm for diagnosing abnormal heart conditions, such as arrhythmias, by measuring and analyzing changes in bio-signals as shown by heartbeats has been developed. Yildirim et al. [5] developed an arrhythmia detection algorithm by applying a convolutional neural network model and laying an electrocardiogram (ECG) signal representing a change in the potential of the myocardium. Bharti et al. [6] developed an efficient algorithm to predict heart diseases by applying various machine learning and deep-learning-based techniques using ECG signals. Sraitih et al. [7] developed an algorithm using various machine learning techniques to automatically diagnose arrhythmias from ECG data. As mentioned earlier, although many studies related to ECG-based monitoring of cardiac activity are underway, the acquisition of ECG signals in daily life is challenging, because general ECG data collection requires that the subjects not move in a lying position, after attaching electrodes and suction cups to their body. Accordingly, a data collection method and an algorithm that can analyze the cardiac activity state using a wearable device instead of the ECG method are required in a mobile environment.

In mobile environments, various wearable sensors based on two signals, PCG and PPG, to analyze cardiac activity have been developed. PCG refers to the recording of blood vibrations generated from the heart and blood vessel walls owing to myocardial contraction, valve closure, and blood flow changes, with a microphone or smart stethoscope. Conversely, PPG is a signal expressed by irradiating peripheral blood vessels with light, and measuring and recording the amount of light absorbed, exploiting the characteristic that a significant amount of blood is released into the arterial system during each systole of the heart. It can be measured on fingers, toes, and earlobes. Smart stethoscopes have been introduced as commercial wearable devices to measure and record PCG in mobile environments. One example is the eKuore smart stethoscope (eKuore, Valencia, Spain) [8], which was developed for convenient use at home. In addition, the StethoMe Stethoscope (StethoMe, Poznań, Poland) [9], was developed to conveniently measure and observe heart sounds in daily life. PPG-based cardiac activity monitoring can be performed with the Galaxy Watch Active 2 (Samsung Electronics, Seoul, South Korea) [10]. The PPG sensor is a widely used sensor because it can be built into various smartwatches. However, in cardiac activity analysis, methods to eliminate noise from PCG and PPG signals and robust algorithms that can recognize peaks are required, and the development of PCG and PPG processing is underway. 

This study focuses on a heart sound analysis method that can be monitored daily using multimodal signals acquired with wearable devices, as in Figure 1. The proposed dual deterministic model-based heart sound analysis (DDM-HSA) is designed in a parallel structure that can simultaneously measure and process two bio-signals (PCG and PPG signals) related to the heartbeat, enabling more accurate identification of heart sounds. Our proposed DDM-HSA method has the following advantages:A parallel structure algorithm based on multimodal methods improves the accuracy of heart sound detection.The proposed DDM-HSA outperforms existing heart sound detection methods using a single signal.By interpolating the S2 peak using the envelope filtering method, S2 detection accuracy can be improved.

The remainder of this paper is organized as follows. Section 2 introduces the database and presents the proposed DDM-HSA method using PCG and PPG signals. The detection performance of the proposed method is demonstrated and discussed in Section 3. Finally, Section 4 concludes the paper.

## 2. Materials and Methods

This section describes the database and methodology used in this study.

### 2.1. Database

#### 2.1.1. Acquisition System and Protocol

A multi-channel physiological data acquisition system, MP160 (BIOPAC System, Inc., Goleta, CA, USA) with a contact acoustic transducer (TSD108A) and PPG sensor (TSD200C) was used to collect the PPG and PCG signals. The acoustic transducer TSD108A was attached to the surface of the chest to measure heart sounds, as shown in Figure 2a,b. In addition, the PPG sensor TSD200C was attached to the middle finger of the left hand to collect PPG signals, as shown in Figure 3. The sampling rate of the data was set at 2000 Hz.

To obtain quantitative data from the subjects, they were required to perform a static activity (rest), as shown in Figure 2. All tasks were performed for 5 min, and the subjects were asked to rest in a chair. 

#### 2.1.2. Subjects

The study was conducted with twenty healthy subjects (ten males and ten females, mean ± standard deviation age of 27.4 ± 3.2 years), as shown in Table 1. Subjects were asked to avoid caffeine-containing beverages and nicotine for 4 h before the experiments [11]. They were also asked to abstain from alcohol and vigorous exercise one day before each experiment [12].

All subjects reported no history of cardiac or psychiatric disorders. In this study, subjects were excluded from the experiment if they reported the following: (1) a history of previous injury or heart repair; (2) being outside the range of 20 to 35 years old; (3) a history of depression, insomnia, or chronic stress; (4) presence of any medical condition that can hinder the subject from performing the exercise; (5) and pregnancy. 

Each subject was briefed on the purpose of the study. The subjects provided written informed consent before participating in the experimental procedures, and the researchers worked to ensure their safety. Ethical review and approval were waived for this study due to only observational equipment being used to observe the signal from the surface of the subject. This study was conducted without any invasive activation, drug administration, or blood collection. In addition, no vulnerable subjects participated in this study, and no personal identification information was collected.

### 2.2. Noise Reduction

A block diagram illustrating the filtering of PPG and PCG signals is shown in Figure 4a. The MP160 (BIOPAC System, Inc., Goleta, CA, USA) commercial device collected raw data by using a 2 kHz sampling rate and a 16 bit analog-to-digital converter. In particular, noise from PPG and PCG was eliminated through the built-in bandpass filter of the MP160 device. All data were acquired using an MP160 device, as shown in Figure 4b. In particular, the PCG signal filtered through the bandpass filter at 20–200 Hz differed from its raw PCG data, and the waveforms of the heart sound S1 and S2 were observed to be precise.

### 2.3. DDM-HSA

#### 2.3.1. Overview

This study proposes a DDM-HSA through S1 and S2 automatic identification algorithms using vascular transit time (VTT). The proposed DDM-HSA algorithm was designed in a parallel structure to identify heart sounds by simultaneously processing heart sounds and pulse waves through VTT. The overall flowchart of the DDM-HSA proposed in this study is shown in Figure 5a. It is designed to identify S1 through preprocessed heart sounds and to recognize S2 waveforms around the systolic peak of pulse waves using VTT. An example of the graphs corresponding to the signal-processing steps in the flowchart is shown in Figure 5b. A detailed description of the proposed method is provided in this subsection.

#### 2.3.2. PCG Analysis

PCG analysis is the first step in the DDM-HSA method, and it recognizes S1 and S2 for heart sound analysis and is divided into five main steps: transformation into an analytic signal, envelope filtering, Shannon entropy, normalization, and heart sound detection. 

Analytic signal $s_{a}\left(t\right)$ is the nonnegative frequency component of the original signal of s(t) [13]. The analytic signal $s_{a}\left(t\right)$ can be defined as follows: (1)$$s_{a}\left(t\right)=s\left(t\right)+j\hat{s\left(t\right)},$$
where j is the imaginary unit and $\hat{s\left(t\right)}$ is the Hilbert transform of s(t). The real-valued $s_{a}\left(t\right)$ with Fourier transform $S_{a}\left(f\right)$ can be defined as follows: (2)$$S_{a}\left(f\right)=\{\begin{array}{c} \frac{1}{2S\left(f\right)},\;if\left(f>0\right),\\ S\left(f\right),\;if\left(f=0\right),\\ 0,\;if\left(f<0\right) \end{array}$$

Using the aforementioned functions, $S_{a}\left(f\right)$ can be defined as follows: (3)$$S_{a}\left(f\right)=\frac{1}{2}S\left(f\right)+sgn\left(f\right)\frac{1}{2}S\left(f\right),$$
where sgn(f) is a sign function, f is the frequency (that is eliminating the negative frequency component by defining the frequency below 0 as zero), and $s_{a}\left(t\right)$ is an analytic signal. The analytic signal $s_{a}\left(t\right)$ is the inverse Fourier transform of $S_{a}\left(f\right)$ expressed as follows: (4)$$s_{a}\left(t\right)=\tilde{F}\left\{S\left(f\right)\right\}+\tilde{F}\left\{sgn\left(f\right)\right\}*\tilde{F}\left\{S\left(f\right)\right\}=s\left(t\right)+j\left[\frac{1}{\pi t}*s\left(t\right)\right]=s\left(t\right)+j\hat{s\left(t\right)},$$

Because multiplication in the time domain is equivalent to convolution in the frequency domain and vice versa, the inverse Fourier transform of sgn(f) and S(f) is a convolution. Currently, the obtained 1/πt∗s(t) is equivalent to $\hat{s}\left(t\right)$ obtained by performing the Hilbert transform of s(t). The Hilbert transform generates a signal whose phase is delayed (shifted) by 90° (quadrature-phase) from the original signal (in-phase); however, energy remains unchanged, because a phase shift is not related to the energy of the signal; only the amplitude changes [14]. Therefore, it is also called a quadrature filter; therefore, the Hilbert transform is performed on s(t), and an analytic signal $s_{a}\left(t\right)$ can be obtained. If ω > 0, then s(t) is cos(ωt), and its Hilbert transform $\hat{s}\left(t\right)$ is defined as cos(ωt−π/2), that is, sin(ωt). Thus, the Hilbert transform extracts the analytic signal $s_{a}\left(t\right)=s\left(t\right)+j\hat{s}\left(t\right)$. Equation (4) is converted to Euler’s formula, and we obtain the following: (5)$$s_{a}\left(t\right)=\cos\left(\omega t\right)+jsin\left(\omega t\right)=A\left(t\right)\cdot \cos\left(\omega t\right)+A\left(t\right)\cdot jsin\left(\omega t\right)=e^{j\omega t},$$
where A(t) and cos(ωt) are the instantaneous amplitude and phase, respectively, obtained by converting the original signal comprising real and imaginary signals. The envelope (energy) and phasor can be extracted.

Envelope filtering (EF) is the task of making the S1 and S2 peaks of the PCG signal more distinct. The heart sound is measured using a microphone or an acoustic device. The noise factor is high, making heart sound identification challenging. This is a critical factor that influences the recognition performance of heart sound identification. In addition, the amplitude of S2 is relatively lower than that of S1. There is a problem, because S2 is regarded as noise (Figure 6). Therefore, we additionally implemented EF to enhance the desired heart sound peaks (S1 and S2) from PCG signals. To calculate the EF of the PCG, we modified the original SEF method proposed in [15] and redefined the EF method for PCG processing.

First, the envelope m(t) and phase of the signals cosϕ(t) were extracted from the real signal s(t). The envelope m(t) was filtered using a bandpass filter (cut-off frequency of 20–200 Hz) to filter the S1 and S2 peaks. The filtered envelope is denoted as $m_{f}\left(t\right)$. Here, we determine the threshold to adjust the elevation of the S2 waveform as follows: (6)$$Threshold\;\left(t\right)=m_{filt}\left(t\right)+\delta \cdot \bar{m_{filt}}$$
where $\bar{m_{filt}}$ and $\bar{\delta }$ represent the mean of the filtered envelope $m_{f}\left(t\right)$ and high-handed heuristic value, respectively. Hence, if the filtered signal $m_{f}\left(t\right)$ is higher than the threshold, the filtered signal $m_{f}\left(t\right)$ is converted by the amplitude of S1, and the final envelope filtered signal z(t) is obtained by multiplying $m_{f}\left(t\right)$ by cosϕ(t). 

Shannon Entropy (SE) calculation is a task that enhances the peak of the EF signal. To detect heart sound waveforms, the location of S1, S2, and their peaks must be identified. This study used the Savitzky–Golay filter, because it has been proven to be more advantageous in preserving peaks compared with other envelope methods [16,17]. The Shannon entropy SE(t) can be obtained as follows: (7)$$SE\left(t\right)=-\left(\hat{SD\left(t\right)}\right)log\left(\hat{SD\left(t\right)}\right).$$

SE(t) was smoothed using a Savitzky–Golay filter to cover the signal SD(t). The filter window size was 900, which is approximately in the interval between the S1 and S2 waveforms. In addition, the degree of the Savitzky–Golay filter was selected as 3, and these parameters were all determined based on empirical and heuristic validation using preliminary data.

Normalization refers to the process of making the value of the filtered envelope scale from 0 to 1 using a min–max method.

S1 and S2 start-point detection refers to the step of primarily detecting the peaks of S1 and S2 in the PCG signals. The normalized signal is converted into an impulse signal as follows: (8)$$Impulse\left(t\right)\{\begin{array}{c} 1,\;\;\;\;\;\;\sigma \times \bar{NSE\left(t\right)}<NSE\left(t\right)\\ 0,\;\;\;\;\;\;\;\;Otherwise \end{array}.$$
where σ and NSE indicate the constant value (added to the mean value of SE(t)) and the normalized SE, respectively. An alpha value of 5 was adopted, using the value set in [18] to determine the threshold for heart sound identification. This was expressed as an impulse signal, and the heart sound time points were extracted. Here, the S2 peak was more accurately detected using the blood VTT based on PPG, which is presented in the next subsection.

#### 2.3.3. PPG Analysis

The heart sound S2, measured during the heart’s diastole, was detected near the systolic peak of the pulse wave when observing the heart sound and the pulse wave. This is due to the fact that a considerable amount of time (called the VTT) is required for blood to flow from the heart to other body parts [19,20,21]. Generally, heart sound S1 is transmitted from the heart to the peripheral blood vessels of the finger after approximately 500 ms. On the other hand, heart sound S2 occurs at approximately 20 ms, at which the systolic peak of the pulse wave is detected [22]. Therefore, PPG was additionally used to precisely recognize S2 obtained primarily from PCG and to revalidate the peak of S2 measured within 20 ms from the systolic peak of PPG. PPG analysis is the second main step of the DDM-HSA method and serves to recognize S2 more precisely.

#### 2.3.4. VTT Calculation and S2 Detection

In order to generate the final DDM-HSA model to improve the accuracy of S1 and S2, the formula (9) can be defined using the heart sound points, the systolic peak points, and the VTT.
(9)$$HS_{sp}=\{\begin{array}{l} S1\;\;\;\;\;\;\;\;\;\;\;\;\;\;\;\left(T_{ppg\_peak}-VTT\leq \;T_{pcg\_sp}\leq \;T_{ppg\_peak}-\beta \right)\\ S2\;\;\;\;\;\;\;\;\;\;\;\;\;\;\;\;\;\;\;\;\;\;\;\;\;\;\;\;\;\;\;\;\;\;\;\;\;\;\;\;\;\;\;\;\left(otherwise\right)\;\;\;\;\;\;\;\;\;\;\;\;\;\;\;\;\;\;\;\;\;\;\;\;\;\;\;\;\;\;\;\;\; \end{array}$$
where $HS_{sp}$ is the time points S1 and S2; $T_{ppg\_peak}$ is the time at which the systolic peak of the pulse and $T_{pcg\_sp}$ is the time at which the heart sound time points, respectively, are detected. VTT is the time at which blood pumped out through heart contraction is delivered to the peripheral blood vessels and is typically observed as roughly 200 ms. β was set at 100 ms, assuming that S2 occurs about 300 ms after VTT and S1 occurs [23].

### 2.4. Performance Measures

The proposed heartbeat sound detection method in this study was evaluated using three performance measures: accuracy (ACC), sensitivity (SEN), and specificity (SPE), which were computed from the following four parameters: True positive (TP): actual heartbeat correctly detected as an actual heartbeat.False negative (FN): not heartbeat detected as not heartbeat.True negative (TN): not heartbeat correctly detected as an actual heartbeat.False positive (FP): actual heartbeat detected as not heartbeat.

ACC is the ratio of the correctly predicted observation to the total observations using (10), and SEN is the ratio of the TP correctly detected to the number of true beats by using (11). SPE refers to the ratio of the TN detected to the total of TN and FP by using formula (12).
(10)Accuracy (ACC)=(TP+TN)/(TP+TN+FP+FN)×100.
(11)Sensitivity (SEN)=TP/(TP+FN)×100.
(12)Specificity (SPE)=TN/(TN+FP)×100.

## 3. Results

Three models were generated to evaluate the heart sound analysis algorithm, as listed in Table 2. Model I is the basic model of DDM-HSA, and Model II is a model in which the DDM-HSA method is applied after applying a window to separate the acquired signal for each segment.

### 3.1. Comparison Result of Models

In the case of S1 peak, because the amplitude of the peak was higher than that of S2, and the shape of the peak was clear, S1 outperformed S2 in terms of accuracy in all models. A comparison of the results for the three models is shown in Figure 7. Eventually, Model III had the highest performance, and S1 and S2 showed average accuracy (unit: %) of 95.39 (±2.14) and 92.55 (±3.74), respectively. The proposed basic model of DDM-HSA was applied in Model I, and the detection performances (unit: %) of S1 and S2 were 87.12 (±4.99) and 63.96 (±13.18), respectively. The data segmentation method was applied to the existing DDM-HSA method in Model II, and the detection accuracies (unit: %) of S1 and S2 were 94.56 (±1.79) and 80.6 (±8.07), respectively.

When comparing the average accuracies of the three models, the performance of Model III was the most robust. This is because Model III applied the EF method, and as the S2 peak was interpolated to the S1 peak, the S2 wave became more apparent. Consequently, the average accuracy difference compared with Model II without EF was approximately 11.95%.

### 3.2. Result of Model I (DDM-HSA)

The performances off all subjects of S1 peak detection in Model I are listed in Table 3. The sensitivity (SEN, unit: %), precision (PRE, unit: %), specificity (SPE, unit: %), and accuracy (ACC, unit: %) of S1 detection were 92.25 (±5.07), 84.05 (±6.43), 81.98 (±8.24), and 87.12 (±4.99), respectively.

As shown in Table 4, the S2 detection performance of Model I was significantly lower than that of S1. This is because the amplitude of the S2 peak needs clarification compared with that of S1; thus, the S1 peak interfered with the S2 peak. In summary, when Model I was applied, the detection performance of S2 was 56.45 (±19.68), 73.83 (±14.47), 72.49 (±12.52), and 63.96 (±13.18) in SEN, PRE, SPE, and ACC, respectively.

### 3.3. Effect of the Window (Model II)

The performance of S1 and S2 to which Model II is applied can be seen in Table 5 and Table 6 to analyze the results of the effect of the data segment. In data segmentation, a particular window is applied to the entire signal to subdivide the data. In the case of S1 (Table 5), the performance of Model II in terms of SEN, PRE, SPE, and ACC was 97.27 (±1.93), 92.37 (±3.03), 91.85 (±3.50), and 94.56 (±1.79), respectively. In the case of S2, SEN, PRE, SPE, and ACC of Model II were 78.39 (±9.14), 88.68 (±6.64), 83.22 (±8.45), and 80.60 (±8.07), respectively. The average accuracy of Model II was improved by approximately 16.64% compared with that of Model I. 

The accuracy of Model I differs from that of Model II because when the entire signal is detected simultaneously, the amplitudes of the peaks differ; a peak with a low amplitude is regarded by other factors as noise. In summary, when a heart sound is detected after a data segment by applying a window to the overall acquired signal, the detection of peaks can be prevented from interfering with each other.

### 3.4. Effect of the Envelope Filtering (Model III)

The indicators of the overall performance of Model III are listed in Table 7 and Table 8. Model III is a DDM-HSA model in which envelope filtering is applied. In summary, the S1 detection performance was 96.88 (±2.79), 94.14 (±2.67), 93.90 (±2.88), and 95.39 (±2.14) in terms of SEN, PRE, SPE, and ACC, respectively. In Model III, detection accuracy was improved by about 8.27 and 0.83 compared to that of Model I and Model II, and there was no significant difference from Model II. Since the envelope filtering is a method of interpolating the relatively low amplitude of S2 according to the amplitude of S1, the performance of S1 did not have a significant effect on envelope filtering.

On the other hand, the performance of S2 in Table 8 was 93.08, 95.12, 90.59, and 92.48 in terms of SEN, PRE, SPE, and ACC, respectively, showing the best performance among the models. This is because Model III, to which the envelope filtering technique is applied, improves the detection performance of S2 by interpolating the amplitude of S2 as much as that of S1.

## 4. Discussion

This paper proposes a novel DDM-HSA method for heart sound analysis in PCG and PPG signals. Our database’s reliable and promising performance demonstrates that the proposed method achieves robust heart sound detection. However, some issues remain to be discussed. In addition, we will discuss the performance of the proposed method compared to other approaches.

### 4.1. Comparison with Other Approaches for Heart Sounds Analysis

Several studies have proposed a novel approach for heart sounds analysis and are summarized in Table 9. Although many studies have been conducted, it is difficult to directly compare the methods with our work because the database used to validate the algorithms is totally different. Therefore, in this subsection comparing other approaches, we focus on and discuss the key contributions of each study. 

The study by Giordano et al. [18] presented a method with robust performance (S1: 99.6%, S2: 98.9%) by measuring the timing of heart sound components in ECG and PCG signals. The most significant difference between the study in [18] and our study is the use of ECG signals. In the study in [18], the ECG signal was used to enhance the performance of heart sound analysis. However, since the ultimate goal of our research is heart sound analysis for the mobile environment, we used PPG, which is data that can be obtained using a wearable device. 

On the other hand, the study by Babu et al. [24] showed a robust performance of heart sound analysis (S1: 100%, S2: 100%) using PPG and PCG signals. They used a PCG but with a microphone condenser attached to the stethoscope’s head. In our study, the most significant difference is that the microphone was attached to the clothes, so that the user could measure the heart sounds without being constrained, as much as possible. The study by Babu et al. [24] showed successful performance because they pursued robust and reliable performance of heart sound analysis.

Finally, the study by Huang [25] focused on ‘Liveness Detection,’ collecting gyroscopes and acoustic signals. Although they also analyzed heart sounds, the results from S1 and S2 were not reported, and it is not easy to make a direct comparison with our study because they used different data from the signals used in our study. However, the study by Huang [25] successfully implemented liveness detection using deep-breath recognition and various feature extraction methods.

### 4.2. Limitations

We presented a novel approach to detect heart sounds S1 and S2 using our database collected from 20 subjects. Our results show promising performance using only PPG and PCG signals for mobile environments. However, there is a limitation in that our data were collected in a controlled laboratory environment, and it is still not easy to generalize our results, because our data need to be more comprehensive. Therefore, the proposed method should be improved based on a larger group of samples for practical use for patients or subjects needing heart sound analysis.

Although we conducted a test experiment of a heart sound analysis algorithm operated in a mobile environment in this paper, the ultimate primary purpose of the proposed algorithm is to apply it to the hardware developed in our preliminary study as Figure 8 [1]. The proposed algorithm was tested based on the data collected from a few subjects to be applied to the smart stethoscope. In addition, we will collect more data in a real-time environment with algorithms applied to smart stethoscopes in the future and generalize the results to overcome the limitations of this paper.

## 5. Conclusions

The study provided a novel approach to improve the technology for detecting heart sounds and analyzing cardiac activity using only bio-signals that can be measured through wearable devices in a mobile environment. The contributions of this study can be summarized as follows:We contributed to the analysis of heart sounds in daily life by presenting a DDM-HSA that can utilize PPG and PCG, which that can be measured using wearable devices.We proposed an envelope filtering method to improve the performance of S2 detection. By applying it to DDM-HSA (Model III), the performance of S2 improved by about 28.59% compared with that of the existing method (Model I).

For the scalability of the proposed approach, several well-defined studies must be considered in the future. As a future research direction, the proposed method should be considered for applications in wearable devices. In our previous studies [1,26], we developed a wearable device that can measure PCG and PPG. The proposed DDM-HSA method will be applied to the device, and clinical trials will be conducted for performance evaluation. This future study is expected to have a significant impact on the analysis of cardiovascular activities in mobile environments.

## Figures and Tables

**Figure 1 sensors-23-02423-f001:**
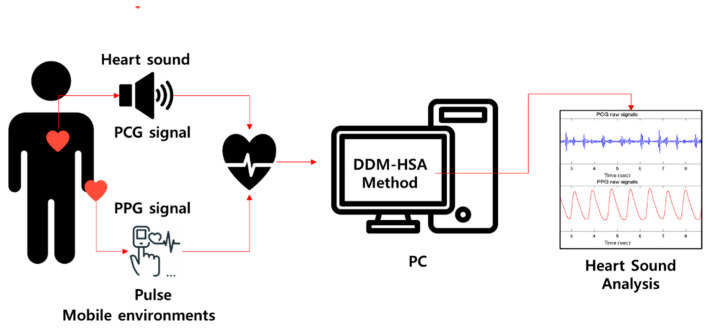
The overall conceptual picture of the method proposed in this paper.

**Figure 2 sensors-23-02423-f002:**
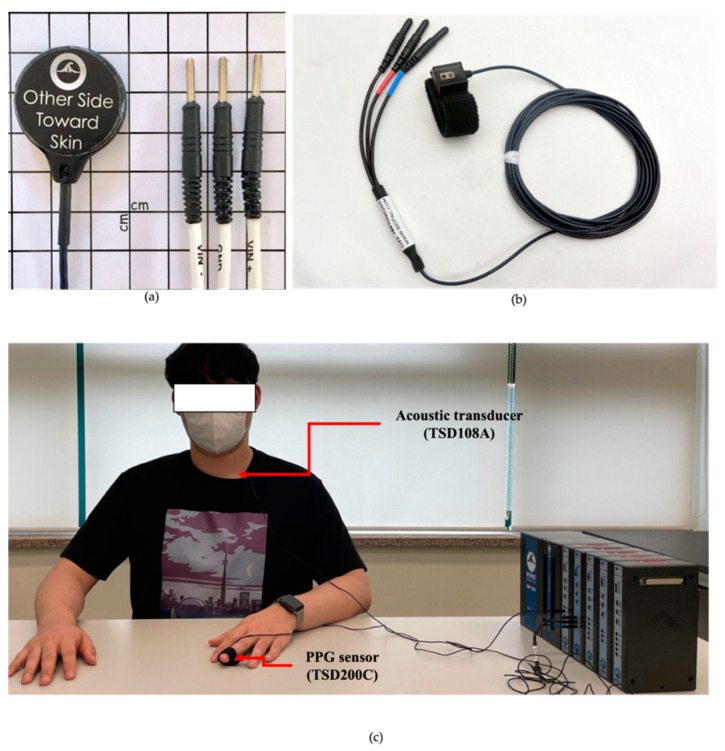
Pictures of experiment for data acquisition: (**a**) Acoustic transducer TSD108A (BIOPAC System, Inc., Goleta, CA, USA); (**b**) PPG sensor TSD200C (BIOPAC System, Inc., Goleta, CA, USA); (**c**) Experimental environment.

**Figure 3 sensors-23-02423-f003:**
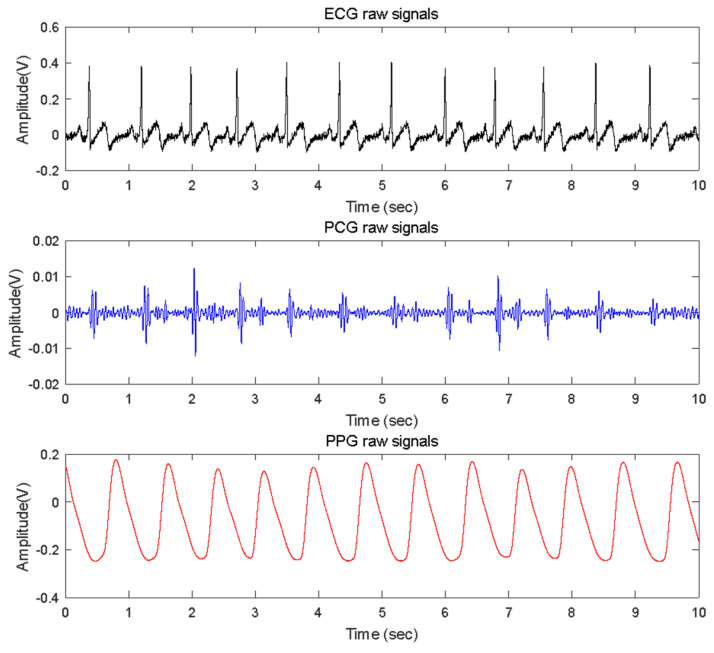
Example of plots of simultaneously recorded ECG, PCG, and PPG signals. The sample data in this plot is for Subject ID XY008, included in the Keimyung University-Heart Sounds Database (KMU-HSDB). The partially extracted data for the graph is from 20,000 to 40,999 of the original data based on the sample number, and the data length is 10 s (sampling rate is 2000 Hz). The top black, middle blue, and bottom red plots represent the raw data of ECG, PCG, and PPG, respectively. The x-axis represents time, measured in seconds; the y-axis is the amplitude of each signal, and the unit is volts (V).

**Figure 4 sensors-23-02423-f004:**
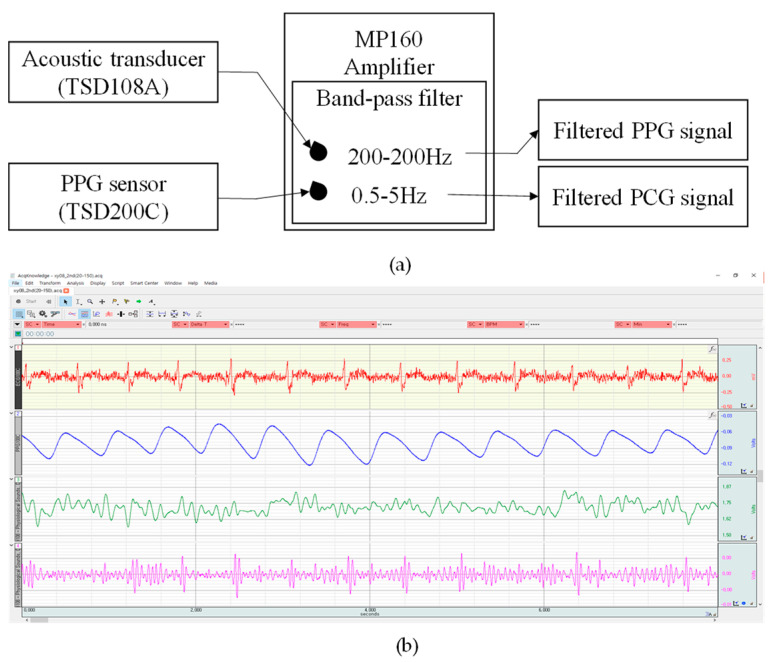
Noise reduction: (**a**) block diagram for filtering (**b**) Example of data obtained through MP160 (BIOPAC System, Inc., Goleta, CA, USA) (Subject ID: XY08). The red, blue, green, and magenta solid lines illustrate the ECG, PPG, raw PCG, and filtered PCG signals (bandpass filter with 20–150 Hz), respectively.

**Figure 5 sensors-23-02423-f005:**
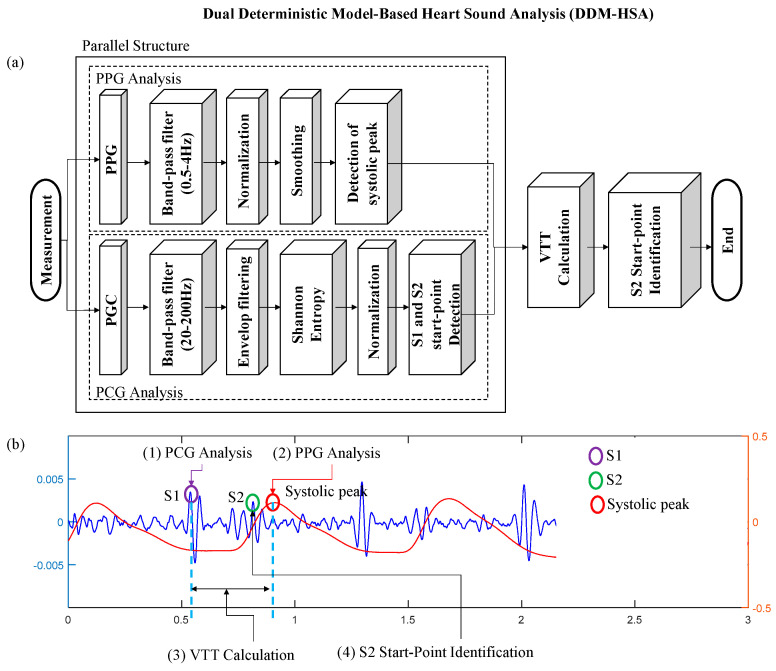
The proposed method of DDM-HSA for daily life monitoring: (**a**) flowchart (**b**) processing example (blue-solid line: PCG; red-solid line: PPG).

**Figure 6 sensors-23-02423-f006:**
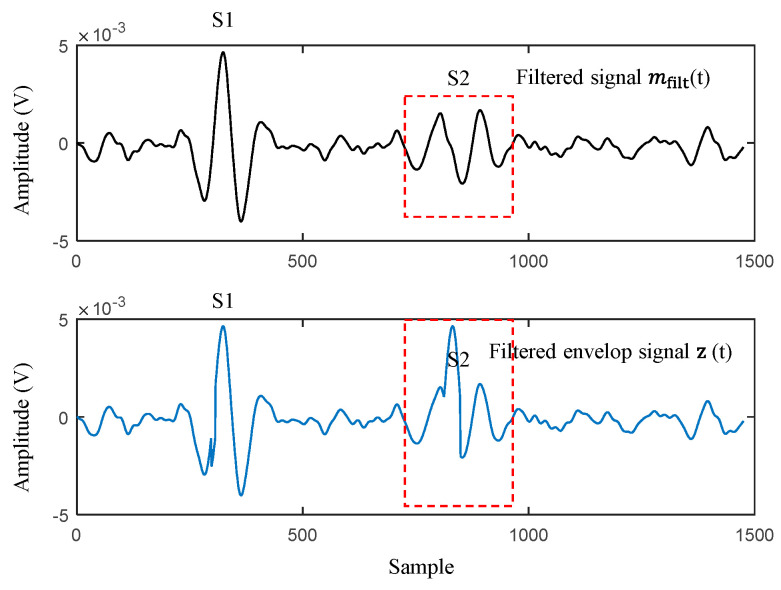
Example of comparison of the filtered signal $m_{filt}\left(t\right)$ and the filtered envelope signal z(t).

**Figure 7 sensors-23-02423-f007:**
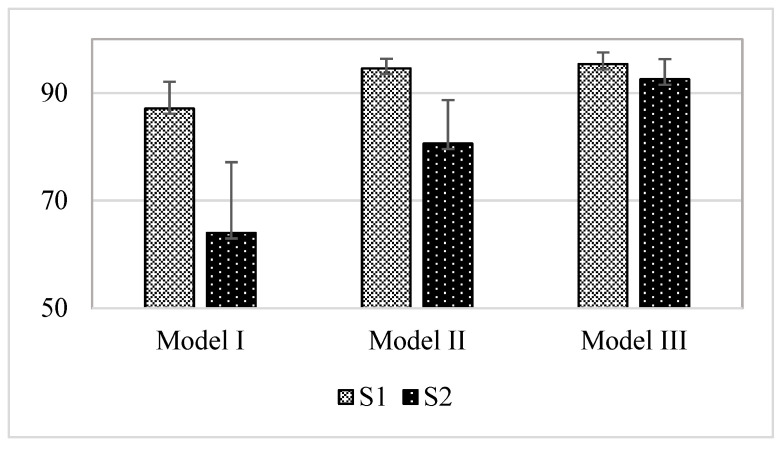
Performance comparison of Model I, Model II, and Model III.

**Figure 8 sensors-23-02423-f008:**
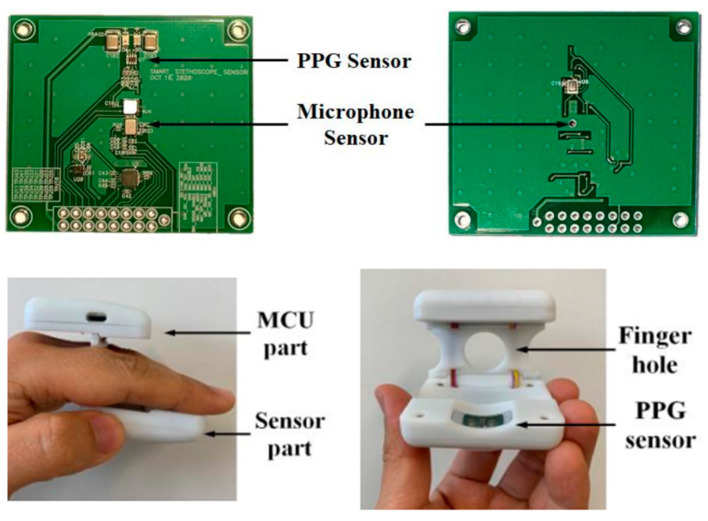
Our preliminary research [1] of hardware development for heart sound analysis in mobile environments. Reprinted/adapted with permission from Ref. [1]. 2021, Sensors.

**Table 1 sensors-23-02423-t001:** Description of subject participating in this study.

Subject ID	Sex	Age
S1	F	32
S2	F	24
S3	F	31
S4	F	28
S5	F	26
S6	F	30
S7	F	28
S8	F	28
S9	F	28
S10	M	29
S11	M	30
S12	M	28
S13	M	34
S14	M	24
S15	M	25
S16	M	31
S17	M	26
S18	M	25
S19	M	27
S20	M	22
TOTAL	M = 10, F = 10	27.4 (3.2)

The numbers in parentheses indicate standard deviations.

**Table 2 sensors-23-02423-t002:** Description of the Model.

Estimation Models	Description
Model I	DDM-HSA
Model II	DDM-HSA + window
Model III	DDM-HSA + window + envelope filter (EF)

**Table 3 sensors-23-02423-t003:** Performance of the S1 peak detection in Model I.

Sub. No.	Mean HR [bpm]	TP	TN	FP	FN	SEN	PRE	SPE	ACC
1	61.50	353.00	255.00	114.00	16.00	95.66	75.59	69.11	82.38
2	77.50	453.00	385.00	80.00	12.00	97.42	84.99	82.80	90.11
3	76.67	386.00	322.00	138.00	74.00	83.91	73.66	70.00	76.96
4	77.17	376.00	363.00	100.00	87.00	81.21	78.99	78.40	79.81
5	73.33	419.00	366.00	74.00	21.00	95.23	84.99	83.18	89.20
6	70.50	355.00	300.00	123.00	68.00	83.92	74.27	70.92	77.42
7	59.67	342.00	318.00	40.00	16.00	95.53	89.53	88.83	92.18
8	72.33	411.00	379.00	55.00	23.00	94.70	88.20	87.33	91.01
9	68.33	388.00	315.00	95.00	22.00	94.63	80.33	76.83	85.73
10	59.17	343.00	251.00	104.00	12.00	96.62	76.73	70.70	83.66
11	61.17	318.00	300.00	67.00	49.00	86.65	82.60	81.74	84.20
12	67.33	375.00	362.00	42.00	29.00	92.82	89.93	89.60	91.21
13	63.17	355.00	315.00	64.00	24.00	93.67	84.73	83.11	88.39
14	67.00	399.00	289.00	113.00	3.00	99.25	77.93	71.89	85.57
15	73.00	414.00	367.00	71.00	24.00	94.52	85.36	83.79	89.16
16	63.67	355.00	327.00	55.00	27.00	92.93	86.59	85.60	89.27
17	86.67	488.00	486.00	34.00	32.00	93.85	93.49	93.46	93.65
18	76.67	422.00	438.00	22.00	38.00	91.74	95.05	95.22	93.48
19	80.17	413.00	448.00	33.00	68.00	85.86	92.60	93.14	89.50
20	62.33	355.00	314.00	60.00	19.00	94.92	85.54	83.96	89.44
AVG	70.00	386.00	345.00	74.20	33.20	92.25	84.05	81.98	87.12
STD	7.75	41.77	61.52	33.18	23.47	5.07	6.43	8.24	4.99

HR indicates the heart rate. TP, TN, FP, and FN mean the true positive, true negative, false positive, and false negative, respectively. SEN, PRE, SPE, and ACC indicate the sensitivity, precision, specificity, and accuracy, respectively.

**Table 4 sensors-23-02423-t004:** Performance of the S2 peak detection in Model I.

Sub. No.	Mean HR [bpm]	TP	TN	FP	FN	SEN	PRE	SPE	ACC
1	61.50	310	268	27	59	84.01	91.99	90.85	87.05
2	77.50	226	187	161	239	48.60	58.40	53.74	50.80
3	76.67	251	199	111	209	54.57	69.34	64.19	58.44
4	77.17	267	243	131	196	57.67	67.09	64.97	60.93
5	73.33	386	300	40	54	87.73	90.61	88.24	87.95
6	70.50	281	138	58	142	66.43	82.89	70.41	67.69
7	59.67	260	111	43	98	72.63	85.81	72.08	72.46
8	72.33	226	188	99	208	52.07	69.54	65.51	57.42
9	68.33	109	311	58	301	26.59	65.27	84.28	53.92
10	59.17	170	297	24	185	47.89	87.63	92.52	69.08
11	61.17	252	50	57	115	68.66	81.55	46.73	63.71
12	67.33	83	243	95	321	20.54	46.63	71.89	43.94
13	63.17	105	211	101	274	27.70	50.97	67.63	45.73
14	67.00	194	298	56	208	48.26	77.60	84.18	65.08
15	73.00	252	136	94	186	57.53	72.83	59.13	58.08
16	63.67	190	241	93	192	49.74	67.14	72.16	60.20
17	86.67	424	255	59	96	81.54	87.78	81.21	81.41
18	76.67	375	168	43	85	81.52	89.71	79.62	80.92
19	80.17	299	213	54	182	62.16	84.70	79.78	68.45
20	62.33	124	197	128	250	33.16	49.21	60.62	45.92
AVG	70.00	239.20	212.70	76.60	180.00	56.45	73.83	72.49	63.96
STD	7.75	94.14	69.26	37.74	77.33	19.68	14.47	12.52	13.18

HR indicates the heart rate. TP, TN, FP, and FN represent true positive, true negative, false positive, and false negative, respectively. SEN, PRE, SPE, and ACC represent the sensitivity, precision, specificity, and accuracy, respectively.

**Table 5 sensors-23-02423-t005:** Performance of the S1 peak detection in Model II.

Sub. No.	Mean HR [bpm]	TP	TN	FP	FN	SEN	PRE	SPE	ACC
1	61.50	366	310	59	3	99.19	86.12	84.01	91.60
2	77.50	451	427	38	14	96.99	92.23	91.83	94.41
3	76.67	434	403	57	26	94.35	88.39	87.61	90.98
4	77.17	424	420	43	39	91.58	90.79	90.71	91.14
5	73.33	431	412	28	9	97.95	93.90	93.64	95.80
6	70.50	411	368	55	12	97.16	88.20	87.00	92.08
7	59.67	350	332	26	8	97.77	93.09	92.74	95.25
8	72.33	422	401	33	12	97.24	92.75	92.40	94.82
9	68.33	404	367	43	6	98.54	90.38	89.51	94.02
10	59.17	351	317	38	4	98.87	90.23	89.30	94.08
11	61.17	363	340	27	4	98.91	93.08	92.64	95.78
12	67.33	388	386	18	16	96.04	95.57	95.54	95.79
13	63.17	371	346	33	8	97.89	91.83	91.29	94.59
14	67.00	400	359	43	2	99.50	90.29	89.30	94.40
15	73.00	434	400	38	4	99.09	91.95	91.32	95.21
16	63.67	377	357	25	5	98.69	93.78	93.46	96.07
17	86.67	500	507	13	20	96.15	97.47	97.50	96.83
18	76.67	438	449	11	22	95.22	97.55	97.61	96.41
19	80.17	464	464	17	17	96.47	96.47	96.47	96.47
20	62.33	366	348	26	8	97.86	93.37	93.05	95.45
AVG	70.00	407.25	385.65	33.55	11.95	97.27	92.37	91.85	94.56
STD	7.75	40.82	51.19	13.94	9.32	1.93	3.03	3.50	1.79

HR indicates the heart rate. TP, TN, FP, and FN represent the true positive, true negative, false positive, and false negative, respectively. SEN, PRE, SPE, and ACC represent the sensitivity, precision, specificity, and accuracy, respectively.

**Table 6 sensors-23-02423-t006:** Performance of the S2 peak detection in Model II.

Sub. No.	Mean HR [bpm]	TP	TN	FP	FN	SEN	PRE	SPE	ACC
1	61.50	333	268	11	36	90.24	96.80	96.06	92.75
2	77.50	400	187	26	65	86.02	93.90	87.79	86.58
3	76.67	388	199	43	72	84.35	90.02	82.23	83.62
4	77.17	300	243	86	163	64.79	77.72	73.86	68.56
5	73.33	411	300	11	29	93.41	97.39	96.46	94.67
6	70.50	355	138	36	68	83.92	90.79	79.31	82.58
7	59.67	311	111	16	47	86.87	95.11	87.40	87.01
8	72.33	343	188	34	91	79.03	90.98	84.68	80.95
9	68.33	352	311	24	58	85.85	93.62	92.84	88.99
10	59.17	264	297	24	91	74.37	91.67	92.52	82.99
11	61.17	311	50	16	56	84.74	95.11	75.76	83.37
12	67.33	274	243	55	130	67.82	83.28	81.54	73.65
13	63.17	255	211	60	124	67.28	80.95	77.86	71.69
14	67.00	294	298	56	108	73.13	84.00	84.18	78.31
15	73.00	294	136	88	144	67.12	76.96	60.71	64.95
16	63.67	264	241	78	118	69.11	77.19	75.55	72.04
17	86.67	434	255	54	86	83.46	88.93	82.52	83.11
18	76.67	399	168	23	61	86.74	94.55	87.96	87.10
19	80.17	344	213	54	137	71.52	86.43	79.78	74.47
20	62.33	254	197	34	120	67.91	88.19	85.28	74.55
AVG	70.00	329.00	212.70	41.45	90.20	78.39	88.68	83.22	80.60
STD	7.75	55.88	69.26	24.09	38.51	9.14	6.64	8.45	8.07

HR indicates the heart rate. TP, TN, FP, and FN represent the true positive, true negative, false positive, and false negative, respectively. SEN, PRE, SPE, and ACC represent sensitivity, precision, specificity, and accuracy, respectively.

**Table 7 sensors-23-02423-t007:** Performance of S1 peak detection in Model III.

Sub. No.	Mean HR [bpm]	TP	TN	FP	FN	SEN	PRE	SPE	ACC
1	61.50	359	349	20	10	97.29	94.72	94.58	95.93
2	77.50	457	464	1	8	98.28	99.78	99.78	99.03
3	76.67	439	427	33	21	95.43	93.01	92.83	94.13
4	77.17	433	425	38	30	93.52	91.93	91.79	92.66
5	73.33	413	422	18	27	93.86	95.82	95.91	94.89
6	70.50	423	386	37	0	100.00	91.96	91.25	95.63
7	59.67	323	336	22	35	90.22	93.62	93.85	92.04
8	72.33	433	403	31	1	99.77	93.32	92.86	96.31
9	68.33	384	370	40	26	93.66	90.57	90.24	91.95
10	59.17	336	319	36	19	94.65	90.32	89.86	92.25
11	61.17	361	347	20	6	98.37	94.75	94.55	96.46
12	67.33	401	388	16	3	99.26	96.16	96.04	97.65
13	63.17	362	342	37	17	95.51	90.73	90.24	92.88
14	67.00	401	360	42	1	99.75	90.52	89.55	94.65
15	73.00	432	419	19	6	98.63	95.79	95.66	97.15
16	63.67	376	360	22	6	98.43	94.47	94.24	96.34
17	86.67	506	511	9	14	97.31	98.25	98.27	97.79
18	76.67	434	446	14	26	94.35	96.88	96.96	95.65
19	80.17	478	463	18	3	99.38	96.37	96.26	97.82
20	62.33	374	349	25	0	100.00	93.73	93.32	96.66
AVG	70.00	406.25	394.30	24.90	12.95	96.88	94.14	93.90	95.39
STD	7.75	47.32	51.57	11.30	11.31	2.79	2.67	2.88	2.14

HR indicates the heart rate. TP, TN, FP, and FN represent true positive, true negative, false positive, and false negative, respectively. SEN, PRE, SPE, and ACC represent sensitivity, precision, specificity, and accuracy, respectively.

**Table 8 sensors-23-02423-t008:** Performance of the S2 peak detection in Model III.

Sub. No.	Mean HR [bpm]	TP	TN	FP	FN	SEN	PRE	SPE	ACC
1	61.50	365	268	22	4	98.92	94.32	92.41	96.05
2	77.50	445	187	27	20	95.70	94.28	87.38	93.08
3	76.67	446	199	43	14	96.96	91.21	82.23	91.88
4	77.17	457	243	37	6	98.70	92.51	86.79	94.21
5	73.33	411	300	22	29	93.41	94.92	93.17	93.31
6	70.50	365	138	15	58	86.29	96.05	90.20	87.33
7	59.67	345	111	21	13	96.37	94.26	84.09	93.06
8	72.33	346	188	30	88	79.72	92.02	86.24	81.90
9	68.33	369	311	24	41	90.00	93.89	92.84	91.28
10	59.17	305	297	18	50	85.92	94.43	94.29	89.85
11	61.17	366	50	12	1	99.73	96.83	80.65	96.97
12	67.33	374	243	18	30	92.57	95.41	93.10	92.78
13	63.17	355	211	13	24	93.67	96.47	94.20	93.86
14	67.00	365	298	23	37	90.80	94.07	92.83	91.70
15	73.00	394	136	18	44	89.95	95.63	88.31	89.53
16	63.67	351	241	9	31	91.88	97.50	96.40	93.67
17	86.67	519	255	12	1	99.81	97.74	95.51	98.35
18	76.67	423	168	13	37	91.96	97.02	92.82	92.20
19	80.17	432	213	9	49	89.81	97.96	95.95	91.75
20	62.33	372	197	16	2	99.47	95.88	92.49	96.93
AVG	70.00	390.25	212.70	20.10	28.95	93.08	95.12	90.59	92.48
STD	7.75	50.26	69.26	8.95	22.60	5.35	1.89	4.64	3.64

HR indicates the heart rate. TP, TN, FP, and FN represent the true positive, true negative, false positive, and false negative, respectively. SEN, PRE, SPE, and ACC represent sensitivity, precision, specificity, and accuracy, respectively.

**Table 9 sensors-23-02423-t009:** Comparison of the approaches for heart sound analysis.

Publication	Database	Approach	Highest Performance	Key Contributions
Giordano et al. [18]	Own database (ECG and PCG)	Measuring the timing of heart sound components	S1: 99.6 ^(a)^ S2: 98.9 ^(a)^ ^(a)^ Sensitivity	Robust Performance
Babu et al. [24]	Own database (PPG and PCG)	Variational mode decomposition-based heart sound endpoint determination	S1: 100 ^(b)^ S2: 100 ^(b)^ ^(b)^ Accuracy	Robust Performance
Huang et al. [25]	Own database (Gyroscope and Acoustic signal)	Deep-breath detection and various feature extraction methods (duration ratio, amplitude ration, correlation coefficient)	S1: N/R S2: N/R	Liveness detection
This study	Own database (PPG and PCG)	Dual deterministic model with window, envelope filtering	S1: 95.39 ^(c)^ S2: 92.55 ^(c)^ ^(c)^ Accuracy	Mobile Environments

N/R denotes not reported.

## Data Availability

The dataset supporting the conclusions of this article is not available due to privacy and ethical reasons.

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
