# Peer review of "DDM-HSA: Dual Deterministic Model-Based Heart Sound Analysis for Daily Life Monitoring"

_sensors, 2023, doi:10.3390/s23052423_

Round 1

Reviewer 1 Report

The reviewed study focuses on a heart sound analysis method that can be monitored daily using multimodal signals received by wearable devices. The authors proposed a model for the analysis of heart sounds based on a double deterministic structure, which models two biosignals (signals PCG and PPG) associated with the heartbeat, which allows more accurate identification of the heart sound. The study is undoubtedly relevant and significant.

In my opinion, a few small remarks can be corrected.

1. So, table 1 seems superfluous to me, it can be moved to additional materials or removed altogether.

2. Tables 3-8 also do not provide important information for each patient, they can be combined leaving only the average values ​​for each model. Full data can also be included in additional materials.

3. Why is there no comparison with existing models that have already been implemented in practice?

Reviewer 2 Report

The paper written by the following Authors: Qun Wei, Miran Lee, Soomin Lee, Heejoon Park entitled “DDM-HSA: Dual Deterministic Model-Based Heart Sound Analysis for Daily Life Monitoring” presents an interesting study on application of multimodal signals acquired by wearable devices.

Although the paper is interesting, I have some major concerns:

Title

The title reflects the results presented here.

Abstract

The abstract is lacking the aim of the study, short material and methods description as well as an informative conclusion. It should be written in more details.

Material and Methods

There is no information about the statistical analysis. It should be included in the manuscript. How Authors calculated mean errors?

Results

Authors did not discuss presented results in relation to the real data. Moreover, there is no information how presented results may be applied in real patients. It should be commented in more details.

Discussion

1. Authors should refer to the manuscripts from the same filed to confront presented results.

2. Limitation to the study should appear at the end of the manuscript.

Round 2

Reviewer 2 Report

I accept the manuscript in the present form.